# Cross-Linking Modification of Ammonium Polyphosphate via Ionic Exchange and Self-Assembly for Enhancing the Fire Safety Properties of Polypropylene

**DOI:** 10.3390/polym12112761

**Published:** 2020-11-23

**Authors:** Yingtong Pan, Zhonglin Luo, Biaobing Wang

**Affiliations:** Jiangsu Key Laboratory of Environmentally Friendly Polymeric Materials, School of Materials Science and Engineering, Jiangsu Collaborative Innovation Center of Photovoltaic Science and Engineering, Changzhou University, Changzhou 213164, China; pyt15261177360@126.com (Y.P.); zhonglinluo@cczu.edu.cn (Z.L.)

**Keywords:** ionic exchange, self-assembly, modified ammonium polyphosphate, polypropylene, combustion behavior

## Abstract

Modified ammonium polyphosphate (MAPP) was prepared as a novel mono-component intumescent flame retardant (IFR) via the ionic exchange between ammonium polyphosphate (APP) and piperazine sulfonate, which is synthesized by self-assembly using 1-(2-aminoethyl) piperazine (AEP) and *p*-aminobenzene sulfonic acid (ASC) as raw materials. This all-in-one IFR integrating three functional elements (carbon, acid, and gas source) showed more efficient flame retardancy and excellent smoke suppression as well as better mechanical properties than the conventional APP. The incorporation of 22.5 wt.% MAPP into polypropylene (PP) eliminated the melt dripping phenomenon and passed the UL-94 V-0 rating. The results of the cone calorimetry test (CCT) revealed that the release of heat, smoke, and CO is significantly decreased, demonstrating that this novel IFR endows PP with excellent fire safety more effectively. For PP/MAPP composites, a possible IFR mechanism was proposed based on the analysis of the pyrolysis gas and char residues.

## 1. Introduction

With the wide application of polymers in construction, electronics, and other fields, the possibility of polymeric materials being exposed to flames has increased significantly [1]. Most polymeric materials are liable to combust, possessing the characteristics of a high heat release rate [2], therefore, improving the fire resistance of polymeric materials is of great significance for diminishing fire hazards and protecting the safety of people’s lives.

Polypropylene, one of the commercially available general-purpose plastics, has been applied in agriculture, industry automobile manufacturing and film packaging, owing to its low toxicity, excellent mechanical properties, electrical insulation, and chemical resistance [3,4]. However, its further application and development are restricted by its disadvantages, including the inherent flammability, severe dripping behavior, and the release of smoke and toxic gases during combustion [5]. Therefore, it is particularly significant to improve both its flame retardancy and smoke suppression performance simultaneously. Recently, phosphorus-based (9,10-dihydro-9-oxa-10-phosphaphenanthrene-10-oxide, APP), nitrogen-based (melamine phosphate, melamine salt of pentaerythritol phosphate kaolinite), mineral-based (aluminum trihydroxide), and carbon-based (graphene) flame retardants have gradually replaced the halogen-containing flame retardants due to their high efficiency, low smoke, and low toxicity [6,7]. Among them, the intumescent flame retardant (IFR), which is made of an acid source (APP, organic phosphoric acid and phosphate ester), blowing agent (urea, melamine, dicyandiamide and their derivatives) and a carbonizing agent (pentaerythritol and piperazine derivatives) is popular. Tang et al. [8] synthesized two charring agents (named as PT-cluster and PT) with different piperazine/triazine group aggregation structures, which generated excellent intumescent flame retardant effects with APP in PP composites. Although APP can act as both an acid source and a blowing agent, it still fails to achieve a good IFR effect in the case of the usage of APP alone [9]. Many strategies have been developed to improve its flame retardancy. The most important choice is to microencapsulate APP with silane coupling agents [10], polyurethane (PU) [11], melamine-containing polyphosphazene (PZMA) [12], or melamine-formaldehyde (MFT) [13] via in situ polymerization. The microencapsulation modification of APP has improved the flame retardant efficiency of PP more or less, but it is still needs to be compounded with a charring agent for which the preparation procedures are complicated and time-consuming. Therefore, the preparation of a mono-component flame retardant which integrates all three sources of IFR has become a research hotspot. Shao et al. tried to modify APP with various organic amines, such as ethylenediamine (EDA) [14], ethanolamine (ETA) [15], diethylenetriamine (DETA) [16], and piperazine (PA) [17] via ionic exchange to prepare three-in-one flame retardant. This method was facile but the flame retardancy still needed to be improved. Thus, a novel modification technology should be developed to overcome this problem.

Self-assembly technology has been applied to produce substances with specific structures by exploiting the interactions between structural units, such as hydrogen bonding and ionic interactions [4,18,19,20]. Recently, self-assembly technology has been adopted to prepare flame retardants due to its facile preparation method and environmentally friendly. Su et al. [21]. modified APP with melamine-formaldehyde and phytic acid (MF-PA) as building blocks by self-assembly reaction. The results demonstrated that the dispersivity of the modified APP in the matrix has been dramatically improved, and the LOI value of the IFR-PP system reached 35% and the UL-94 V-0 rating was achieved at a loading level of 25 wt.% of modified APP/ CFA (4:1, weight ratio). Jin et al. [22]. prepared a novel macromolecular IFR (AM-APP) via supramolecular reactions between melamine and *p*-aminobenzene sulfonic acid, followed by an ionic exchange with APP. With the incorporation of 22 wt.% AM-APP and 3 wt.% TiO_2_, polyamide 11 composites showed high LOI value, upgraded UL-94 rating, and an 81.2% reduction on pHRR value. Although some progress had been achieved in the modification of APP via self-assembly, it still needs to be matched with charring agents or metal oxides to pass the UL-94 V-0 rating. In addition, few pieces of literature have been reported on the synthesis of mono-component flame retardant through self-assembly technology and ionic exchange.

The current work reported the synthesis and characterization of a mono-component flame retardant prepared by self-assembly reactions between 1-(2-aminoethyl) piperazine and *p*-amino benzenesulfonic acid, followed by an ionic exchange with APP. The facile preparation method was considered environmentally friendly, using anhydrous ethanol and deionized water as the solvent. The flame retardancy, combustion behaviors, and flame retardant mechanism of the IFR-PP system were investigated in detail.

## 2. Materials and Methods

### 2.1. Materials

PP, F401, melt flow rate = 2.5 g (10 min)^−1^ (230 °C, 2.16 kg) was provided by Yangzi Petroleum Chemical Company (Nanjing, China). Ammonium polyphosphate (TY-432, crystalline form II, degree of polymerization > 1000) was purchased from Yunnan Tianyao Chemical Co., Ltd. (Kunming, China). *p*-Aminobenzene sulfonic acid (AR, 99.5%) and 1-(2-aminoethyl) piperazine were supplied by Aladdin Industrial Corporation (Shanghai, China). Absolute ethyl ethanol (AR, 99.5%) was obtained from Sinopharm Group Co., Ltd. (Shanghai, China). Deionized water was self-made. All of the commercial materials were used directly without further purification.

### 2.2. Modification of APP

Figure 1 illustrated the modification process of ammonium polyphosphate involving two stages.

Firstly, the piperazine sulfonate (intermediate) was prepared via self-assembly as shown in stage I. Under stirring, ASC (17.3 g, 0.1 mol) was dissolved in deionized water at 90 °C. Hereafter, AEP (13.1 mL, 0.1 mol) was dropped into the ASC solution at a constant titration rate (2 s / drop). The reaction system was kept 95 °C for 3 h. After vacuum distillation and suction filtration, the straw yellow powder (AEP-ASC) was obtained and then dried at 80 °C in the vacuum oven for 24 h.

Secondly, the modified APP (MAPP) was synthesized via the ionic exchange between intermediate and APP as shown in stage II. To a three-neck round-bottom flask equipped with a stirrer, a thermometer and a reflux condenser were added APP (10 g), 50 mL ethanol, and 20 mL water. The mixture was refluxed under stirring for 12 h. Afterward, the intermediate (10 g) was introduced into the mixture, and the reaction continued for 2 h. Then, the light yellow and viscous solid was obtained from vacuum filtration and washed by deionized water (removing unreacted intermediate) three times. After drying under a high vacuum at 80 °C overnight, the final product (light yellow powder) was obtained and defined as modified APP (MAPP).

### 2.3. Sample Preparation

The IFR/PP blends were obtained by mixing the APP or MAPP with different mass ratios. The specific formulas were shown in Table 1. Then the IFR/PP blends were mixed at 190 °C for 5 min in an internal mixer (US-70C, Changzhou Suyan Technology Co., Ltd., Changzhou, China). The samples for fire behavior characterization were pressed to the sheet at 195 °C for 5 min under 10 MPa of the pressure under a plate vulcanizer (ZHY-W, Chengde Testing Machine Factory, Hebei, China). The samples for mechanical measurements were injection-molded in a miniature injection molding instrument (WZS-10D, Xinshuo Precision Machinery Co., Ltd., Shanghai, China). The melting temperature was 200 °C, the mold temperature was 40 °C, the holding pressure was 0.6 MPa, and the holding time was 10 s.

### 2.4. Characterization

The Fourier transform infrared (FTIR) spectra were scanned by a Perkin Elmer instrument (Waltham, MA, USA) at room temperature. The samples were mixed with KBr pellets including 0.5 mg sample and 50 mg KBr and scanned 32 times over a spectral range of 4000–450 cm^−1^ with a resolution of 4 cm^−1^.

X-ray diffraction (XRD) test was carried out by a power D/MAX2500 diffraction (Rigaku Corporation, Tokyo, Japan) using Cu Ka radiation under a scanning rate of 3°/s from 5° to 60° (2*θ*).

X-ray photoelectron spectroscopy (XPS) was determined by an ESCALAB 250XI system (Thermo Fisher, Waltham, MA, USA).

The microstructures of flame retardants, char residues, and sections were measured using an SEM instrument (Zeiss SUPRA55, Jena, Germany). The specimens were sputter-coated with a conductive gold layer before observation.

The UL-94 flammability classification was measured by a CZF-5 instrument (Shine Ray Instrument Co. Ltd., Nanjing, China) according to ASTM D3801. The dimensions of the samples are 130 mm × 13 mm × 3.2 mm.

Limiting oxygen index (LOI) was measured by an LOI analyzer (JF-3, Jiang Ning Co. Ltd., Nanjing, China). The dimensions of the samples are 130 mm × 6.5 mm × 3.2 mm according to GB/T 2406-93 standard.

Thermogravimetric analysis (TGA) was tested by a Perkin-Elmer TGA 4000 with a heating rate of 10 °C/min under nitrogen and oxygen atmosphere at temperatures ranging from 30 °C to 850 °C. TG-FTIR analysis was carried out under a nitrogen atmosphere at a heating rate of 10 °C/min from 30 °C to 850 °C. The flame retardant samples used in the tests were all fine solid powders with a mass of 8 ± 0.5 mg; the samples of pure PP and its blends used were block solids with a mass of 8 ± 0.5 mg.

The flammability of pure PP and PP blends were tested by a cone calorimeter device (Fire Testing Technology, East Grinstead, UK) according to ISO 5660-1. The dimension of the square sample was 100 mm × 100 mm × 3 mm, and the irradiation power was 35 kW/m^2^.

Raman spectroscopy measurement was carried out with a DXR laser Raman spectrometer (Thermo Scientific,) using a 532 nm helium-neon laser line at room temperature.

Tensile measurements were conducted on a universal tensile testing machine (WDT-5, Shenzhen Kai Qiang Experimental Instrument Co. Ltd., Guangzhou, China) according to GB/T 1040-2006 at a tensile speed of 50 mm/min. The Izod notched impact strength of the specimens was measured with an impact tester (XJU-22, Chengde Testing Machine Co. Ltd., Sichuan, China) according to GB/T 1843-2008. The values of all the mechanical properties were calculated as averages over five specimens.

## 3. Results and Discussion

### 3.1. Characterization of MAPP

#### 3.1.1. FTIR Analysis

Figure 2a illustrates the FTIR spectra of AEP, ASC, and AEP-ASC. The characteristic peaks at 2939 and 2813 cm^−1^ are attributed to the symmetrical and antisymmetric –CH_2_– stretching vibration of AEP, and the peaks at 1315 and 1120 cm^−1^ represent the –S=O stretching vibration of ASC [23]. Noticeably, the peak at 2647 cm^−1^ (S–OH) [24] in the spectrum of ASC disappears and two new peaks at 2446 and 3265 cm^−1^ (NH_2_^+^ and NH_3_^+^ bending vibration) are observed in the spectra of AEP-ASC, which indicates the formation of –NH_2_^+^–O– and –NH_3_^+^–O– [1,25]. Furthermore, the spectrum of AEP-ASC shows sharp peaks at 3469, 3347 and 3232 cm^−1^ owing to amino groups [11]. These results demonstrate that intermediate (AEP-ASC) is obtained by self-assembly. The chemical structures of the APP and MAPP are also explored by FTIR spectra (Figure 2b). Both APP and MAPP display the typical peaks at 1263 cm^−1^ (P=O), 1087 cm^−1^ (P–O symmetric stretching), and 882 cm^−1^ (P–O asymmetric vibration) [26]. As compared with the spectra of APP, however, that of the MAPP displays some new peaks which are ascribed to the absorption of the benzene ring (1600 and 689 cm^−1^, etc.), the stretching vibration of O=S=O (1320 and 1180 cm^−1^) and especially the NH_2_^+^ and NH_3_^+^ bending vibration (2481 and 3248 cm^−1^). The results initially demonstrate that the APP was successfully cross-link modified with AEP-ASC by ionic exchange.

#### 3.1.2. XRD Analysis

XRD patterns of APP and MAPP are presented in Figure 2c. Obviously, the position of the diffraction peaks of APP and MAPP are almost the same, indicating that the crystalline structure of APP is not influenced by cross-linking modification.

#### 3.1.3. XPS Analysis

The XPS spectra of APP and MAPP are shown in Figure 2d. As can be seen, MAPP shows much higher carbon content (44%), lower contents of oxygen (32%), and phosphorus (8%) than APP. Furthermore, a new peak appears at 168.1 eV for MAPP, which is ascribed to the S_2p_ from the intermediate (AEP-ASC). The fitted N_1S_ spectra of APP and MAPP are presented in Figure 2e,f. For APP, the binding energy at around 401.2 eV is assigned to NH_4_^+^, and the peak at 399.3 eV might correspond to the N (–P–NH–P–). This result of APP spectra agrees with Wang’s report [17]. Although the NH_4_^+^ peak still appears for MAPP, three new peaks at 399.7 eV, 400.3 eV, and 400.7 eV are observed [16,17]. It indicates that NH_2_^+^ and NH_3_^+^ are formed and take place of NH_4_^+^ partially. This result further confirms the successful ionic interactions between intermediate and APP.

#### 3.1.4. Surface Morphology and EDS test

SEM is used to investigate the micro-morphology of the APP and MAPP, and the corresponding images are shown in Figure 3. Apparent changes in the surface morphology between APP and MAPP are easily observed. The pure APP particle shows a smooth surface, and its particle size is about 10 μm (Figure 3a,b). However, the MAPP particles look like a cross-linked aggregate of APP particles and display a rough surface (Figure 3c,d). This could be interpreted as due to H-bonding and ion-dipole forces that contribute to particle aggregation and fusion [16]. The changes in the surface morphology of MAPP are consistent with the proposed model as shown in Figure 1. Moreover, the elemental composition of the APP and MAPP is evaluated from the EDS test (Figure 3e). By comparison with APP, the MAPP presents higher content of C (24.27%) and lower content of P (11.53%) as well as the appearance of S element. Both the changes in the surface morphology and element composition between APP and MAPP demonstrate the successful cross-linking modification of APP evidently.

### 3.2. Fire Behavior

#### 3.2.1. Reaction to Small Flame (UL 94 Vertical Burning and LOI)

The flame retardant data of the neat PP and IFR-PP blends are summarized in Table 1. The neat PP shows serious dripping with no rating level during the UL-94 test, and its value of LOI is merely 17%, indicating its extreme flammability. In the case of the incorporation of APP alone, the PP/APP blend fails to pass the UL-94 V-0 test and gives of LOI value of 19% even at its loading level up to 25 wt.%, revealing that APP is not very practical to improve the flame retardancy of PP. However, the LOI values and flame retardant rating of PP/MAPP blends are increased significantly with the incorporation of MAPP. For example, the IFR-PP blend containing 22.5 wt.% MAPP gives 30% of LOI value and UL-94 V-0 rating. It demonstrates that the flame retardant efficiency of MAPP is much higher than that of APP for IFR-PP blends.

#### 3.2.2. Combustion Behavior Under Forced-Flaming Scenario (Cone Calorimetry Test)

The cone calorimetry test (CCT) is currently the ideal test to assess the burning behavior of polymeric materials under an ongoing fire or forced combustion conditions. The characteristic curves including the heat release rate (HRR), total heat release (THR), smoke production rate (SPR), total smoke production (TSP), CO, and CO_2_ production (COP, CO_2_P) are illustrated in Figure 4, and some important parameters such as time to ignition (TTI), the peak of heat release rate (PHRR), THR, and average mass loss rates (AMLR) are listed in Table 2.

HRR is generally the most critical performance parameter to characterize the fire intensity, and PHRR indicates the maximum degree of heat release during combustion. As presented in Table 2, the pure PP gives 718.3 kW/m^2^ of PHRR value and 57.3 MJ/m^2^ of THR, which are reduced to 354.7 kW/m^2^ and 47.9 MJ/m^2^ with the incorporation of 25 wt.% APP, respectively. The corresponding values are further reduced significantly with the substitution of APP with MAPP. For example, the IFR-PP containing 25 wt.% MAPP presents the lowest PHRR (155.9 kW/m^2^) and THR values (44.9 MJ/m^2^), which are reduced by 78.3% and 21.6%, respectively, as compared to those of the neat PP. The sharp decline of the PHRR and THR values indicates that MAPP has a more significant role in fire safety than APP.

Fire growth index (FGI) and fire performance index (FPI) are used to assess the fire hazard of polymers, which can be calculated according to the following equation [27] based on the HRR curves and TTI values:FGI = PHRR/T_PHRR_ FPI = TTI/PHRR

The obtained FGI and FPI values of the neat PP and IFR-PP blends are also listed in Table 2. Generally, the lower FGI and higher FPI values imply higher fire safety properties [28]. As can be seen, the IFR-PP blends display much lower FGI values whilst greater FPI values than that of the pure PP. Noticeably, the FPI value (0.22 m^2^s/kW) of PP/25%MAPP is increased by 57.1% whilst the FGI value (0.72 kW/m^2^s) is decreased by 61.5% as compared with these corresponding values of PP/25%APP (0.14 m^2^s/kW of FPI and 1.87 kW/m^2^s of FGI, respectively). Furthermore, taking the much lower average mass loss rate (AMLR, 0.027 g/s) of PP/25%MAPP than that (0.043 g/s) of PP/25%APP into consideration, it can be concluded that MAPP is superior to APP in fire safety.

It is well known that most victims suffocate to death by inhalation of smoke and CO in fire accidents. As such, smoke suppression is very significant for flame retardant materials. As can be seen in Table 2, the total smoke production (TSP) is reduced from 48.8 m^2^ for pure PP to 6.4 m^2^ for PP/APP while their peaks of the smoke production rate (PSPR) are almost the same. Fascinatingly, both the PSPR and TSP values of PP/25%MAPP are reduced drastically as compared with that of the pure PP or PP/25%MAPP, indicating the superior smoke suppression of MAPP. Moreover, the CO emissions during CCT are ascribed to the insufficient combustion of decomposed volatiles. It is apparent that CO and CO_2_ relaease of IFR/PP composites are weaker than those of pure PP from Figure 4e,f, which was contributed to the formed char at the early stage and barrier performance of the cross-linking structure of MAPP. It is worth noting that the mean CO yeild (0.15 kg/kg) of PP/25%MAPP specimen is half of that of pure PP (0.32 kg/kg) due to the excellent intumescent char residue on the sample surface, implying the possibility of suffocation can be significantly inhibited during evacuation.

### 3.3. Thermal Stability Analysis

Figure 5 and Figure 6 present the TG and DTG curves of the flame retardants and IFR-PP samples under nitrogen and air atmosphere, respectively. Corresponding data, such as the initial thermal decomposition temperature (*T*_5*wt*%_), the temperature at maximal degradation (*T_max_*), Rate of *T_max_*, and char residue at 800 °C are listed in Table 3 and Table 4. It can be observed that the MAPP shows high *T*_5*wt*%_ values, no matter under the nitrogen (303.1 °C) or air (270.9 °C) atmosphere, which fully meets the melt blending temperature of PP (190 °C). While APP presents two primary mass loss processes, ascribing to thermal removal of H_2_O and NH_3_, and crosslinking reactions between phosphorus-containing acid. It is noticeable that thermal decomposition process of MAPP is little different from APP, due to the introduction of AEP-ASC. The deamination, dehydration and decomposition of ASC at around 225–380 °C make the first DTG peak become wider. The second stage after 500 °C corresponds to the pyrolysis of APP chains as well. Furthermore, the pure PP shows a single degradation process with the *T*_5*wt*%_ and *T_max_* at 373.5 °C and 455.4 °C under nitrogen atmosphere. Moreover, there is no char residue left at 800 °C no matter whether under nitrogen or air atmosphere, indicating pure PP fails to form char alone. On the contrary, two-stage degradation is observed for all IFR-PP blends. The first-stage at low temperature is mainly ascribed to the degradation of the IFR and PP, and the second-stage at high temperature is attributed to the decomposition products (polyphosphoric acids, pyrophosphate, and metaphosphate acid) of APP or MAPP [1]. Furthermore, all samples display a lower degradation temperature under the air atmosphere than that under the nitrogen atmosphere owing to thermo-oxidative degradation [24]. As compared with the pure PP, all PP/APP samples display greater *T*_5*wt*%_ values, which is ascribed to the protection of the thick polyphosphoric acid fluid produced by the decomposition of APP [25]. In the case of substitution of APP with MAPP, the *T*_5*wt*%_ values of PP/MAPP samples move to lower temperatures since the *T*_5*wt*%_ of MAPP is lower than that of APP. It is noteworthy that the char residue of PP/25%MAPP is up to 5.4% at 800 °C under the air atmosphere, which is higher than that of pure PP and PP/APP samples. Such an observation demonstrates that the phosphoric acid released by APP can catalyze the carbon formation of the piperazine ring from AEP and the physical carbon formation of the benzene ring in ASC make MAPP itself can form a stable carbon layer, which acts as a barrier between oxygen and PP matrix [11].

### 3.4. Flame-Retardant Mechanism Analysis

#### 3.4.1. Gas Phase Analysis

A TG-FTIR test was performed to analyze the volatile gases released from IFR-PP samples, which helps to explore the flame-retardant mechanism of the gas phase. Figure 7 illustrates the FTIR curves and 3D spectra of the evolved gaseous products during the pyrolysis of pure PP, PP/25%APP, and PP/25%MAPP. All the samples display the following characteristic peaks: alkanes (2960, 2920 cm^−1^), alkenes (1647, 1465, and 1371 cm^−1^) and a diene (890 cm^−1^) [10]. However, the intensity of the abovementioned peaks in the spectra of PP/25%MAPP is weaker than that of the pure PP and PP/25%APP, suggesting that volatile components are effectively suppressed during the degradation process of PP/25%MAPP specimens [21]. As compared with the pure PP, both PP/25%APP and PP/25%MAPP specimens show characteristic peaks ascribed to P=O (1258 cm^−1^), P–O (1087 cm^−1^), and NH_3_ (966 and 937 cm^−1^), which contribute to the thermal degradation of APP and scission of polyphosphoric acids [29]. Furthermore, it is noticeable that a new peak at 1499 cm^−1^ (SO_2_) is found only in the spectrum of the PP/25%MAPP specimen [22,30]. The appearance of these non-combustible gases (NH_3_ and SO_2_) can dilute the released combustible gases (hydrocarbons, etc.).

#### 3.4.2. Analysis of Char Residue

Digital photographs of the residues after CCT are shown in Figure 8. Obviously, there is no residue left for pure PP and a minute quantity of residues for PP/25%APP. This indicates that the incorporation of the unmodified APP alone cannot promote the formation of an expanded char layer. However, more intumescent and expanded char residues with expansion heights of 1.7 cm (Figure 8c’) and 2.5 cm (Figure 8d’) are observed for PP/22.5%MAPP and PP/25%MAPP, respectively. The expansion height is usually adopted to assess the swelling degree or quality of the char layer. Accordingly, it is concluded that MAPP has a greater effect on the formation of the intumescent char layer than the unmodified APP, which can obstruct the heat and mass transfer between the gas and condensed phase.

The microstructure of the residues after CCT was observed further via SEM, and the resultant images are presented in Figure 9. Clearly, numerous large holes are observed both in the outer and inner char residues of PP/25%APP whilst compact and continuous char layer is observed on the surface of PP/MAPP blends. Especially, the appearance of some char bubbles in the PP/25%MAPP surface residues indicates that the incorporation of MAPP enhances the compactness of char residues and thus forms a much excellent IFR system [5]. However, the honeycomb structure still exists in the inner char residue of PP/MAPP samples, which is ascribed to the diffusion of pyrolysis gas.

The elemental compositions of the char residue of the PP/25%MAPP sample after CCT were investigated by the XPS analysis (Figure 10). As can be seen, the main elements in the residue are C, O, P, and N. In N_1S_ spectra (Figure 10b), the peaks at 400.5 eV, 401.2 eV, and 402.3 eV are attributed to P–N–P, NH_4_^+,^ and P–C–N, respectively [21,23]. The O_1S_ peaks (Figure 10c) at 531.1 eV and 532.2 eV are assigned to =O (including P=O and C=O) and –O– (including C–O–C and P–O–C [10]), respectively. In the case of P_2P_ spectra (Figure 10d), the peaks around 133.9 eV and 134.5 eV correspond to P=O and P–O–C structure, respectively [22,31]. The results above demonstrate that the existence of P–C–N and P–O–C structure plays a vital role in forming a dense and continuous char layer.

Raman spectroscopy is a useful tool to characterize the degree of graphitization of char residues. The Raman spectra are mainly divided into G bands (about 1590 cm^−1^, showing the graphitic structure) and D bands (about 1365 cm^−1^, representing lattice defects of carbon atom). Furthermore, the degree of graphitization of the char residue can be assessed by the area ratio of two peaks (A_D_/A_G_). Generally, the lower ratio of A_D_/A_G_ means the higher graphitization degree and better thermal-oxidative stability of the char [32,33]. As shown in Figure 11, the A_D_/A_G_ values are calculated to be 2.57 for PP/25%APP, 2.24 for PP/22.5%MAPP, and 2.12 for PP/25%MAPP, respectively. As such, it is concluded that the incorporation of MAPP facilitates the graphitization to form much dense and continuous surface char layers, which is in agreement with the SEM observation results.

#### 3.4.3. Possible Flame-Retardant Mechanism

Based on the analysis of the volatile gas and char residues, a possible flame-retardant mechanism is proposed and illustrated in Figure 12. In the initial stage of the combustion, the dehydration of MAPP and the cleavage of ionic bonds result to release of some incombustible gases (such as NH_3_, SO_2_, and water vapor) which can fully dilute the oxygen and heat. With rising temperature, MAPP is thermally decomposed into polyphosphoric acid, pyrophosphoric acid, or metaphosphoric, all of which can esterify and crosslink with the char precursor (piperazine ring) [10]. Moreover, the phosphate ester radicals and pyrophosphate radicals released from the decomposition of MAPP readily quench O·, H·, and OH· radicals [34]. Additionally, the benzene ring in the intermediate acts as both the cross-linking agent and crystal core [23]. As such, the char layer containing P–O–C and P–C–N structure is formed and its quality is improved owing to the crosslinking networks. It is noteworthy that although the initially formed char layer did not swell, its continuous and compact structure hinders the gas overflowing and the accumulation of the incombustible gases keeps the char layer expanding. As a result, an intumescent and dense char layer is formed, which protects the underlying substrate from further thermal decomposition. The physical (expansion and morphology) and chemical (thermal stability) properties of the char layer were enhanced after introducing MAPP and the possible IFR mechanism is shown in Figure 12 [21].

### 3.5. Mechanical Properties

Figure 13 illustrates the typical stress-strain curves of pure PP and IFR-PP blends, and the mechanical properties data are provided in Table 5. The pure PP gives 39.1 MPa of tensile strength, 147.1% of elongation at break, and 4.6 kJ × m^−2^ of impact strength, respectively. Although the mechanical properties of IFR-PP blends are decreased to a certain extent by comparison with the pure PP, the PP/25%MAPP sample shows better mechanical properties than the PP/25%APP. Namely, the tensile strength, elongation at break, and impact strength of PP/25%MAPP are increased by 12.3%, 110.5%, and 37.9% as compared with PP/25%APP, respectively. This difference might be related to interface interaction between the dispersion phase (MAPP or APP) and the PP matrix. Figure 14 presents the SEM images of the impact fractured surface of PP, PP/25%APP, and PP/25%MAPP. APP particles fail to adhere to the PP matrix, and the interface between the APP particles and PP matrix is clearly visible (Figure 14b). Conversely, the MAPP particles are uniformly embedded in the PP matrix (Figure 14c), which leads to an improvement of the elongation at break and impact strength. This observation is probably ascribed to that the modification of APP improved the interfacial adhesion, and the addition of MAPP had little effect on the mechanical properties of IFR-PP samples.

## 4. Conclusions

A novel mono-component flame retardant (MAPP) was prepared by the modification of conventional APP with piperazine sulfonate. The MAPP showed more effective flame retardancy and smoke suppression than the traditional APP. The PP/MAPP composite passed the UL-94 V-0 rating and achieved 30% of LOI value at 22.5 wt% loading level of MAPP. The HRR, PHRR, and TSP values of PP/22.5%MAPP were reduced significantly as compared with those of the pure PP or PP/25%MAPP. The results of TGA-FTIR and char residue analysis revealed that some incombustible gases were released and intumescent, dense and continuous char residues were produced with the incorporation of MAPP. The PP/25%MAPP sample also displayed better mechanical properties than the PP/25%APP.

## Figures and Tables

**Figure 1 polymers-12-02761-f001:**
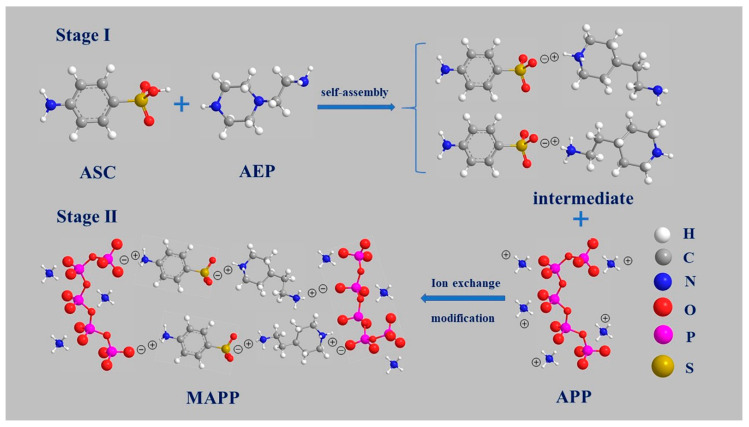
The synthetic route of MAPP.

**Figure 2 polymers-12-02761-f002:**
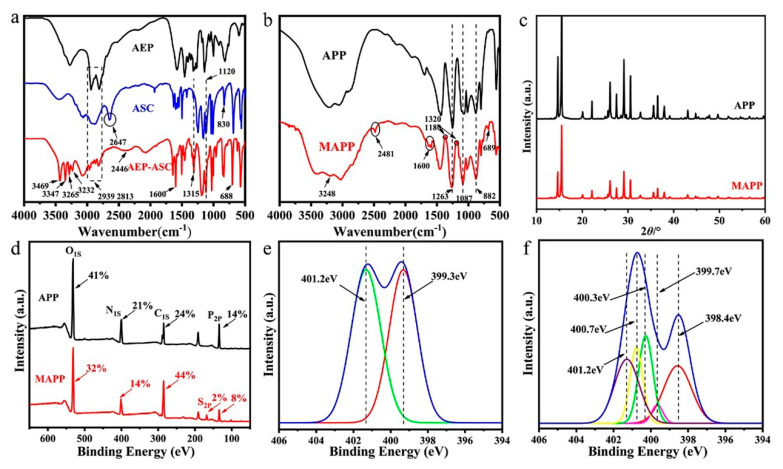
(**a**) FTIR spectra of AEP, ASC and intermediate (AEP-ASC); (**b**) FTIR spectra of APP and MAPP; (**c**) XRD patterns of APP and MAPP; (**d**) XPS survey spectra; N_1_s XPS spectra of (**e**) APP and (**f**) MAPP.

**Figure 3 polymers-12-02761-f003:**
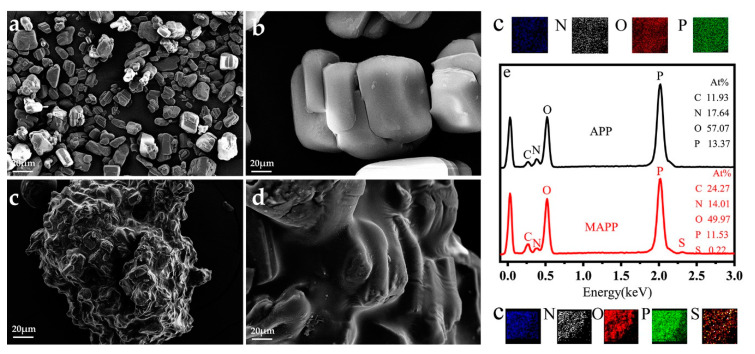
SEM images of (**a**,**b**) APP and (**c**,**d**) MAPP; (**e**) EDS image of APP and MAPP.

**Figure 4 polymers-12-02761-f004:**
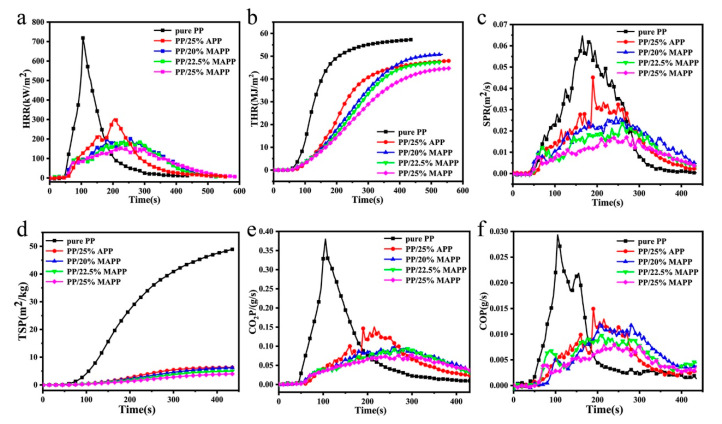
HRR (**a**), THR (**b**), SPR (**c**), TSP (**d**), CO_2_P (**e**), and COP (**f**) curves of samples during the combustion in CCT.

**Figure 5 polymers-12-02761-f005:**
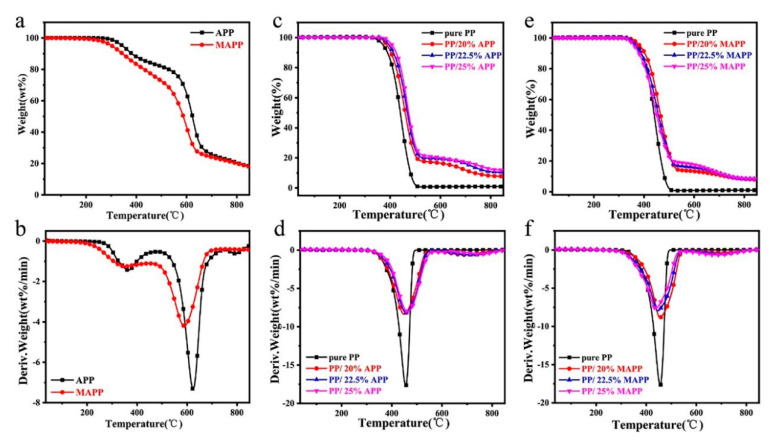
(**a**) TG and (**b**) DTG curves of APP and MAPP. (**c**,**e**) TG and (**d**,**f**) DTG curves of PP and its flame-retardant blends under the nitrogen atmosphere.

**Figure 6 polymers-12-02761-f006:**
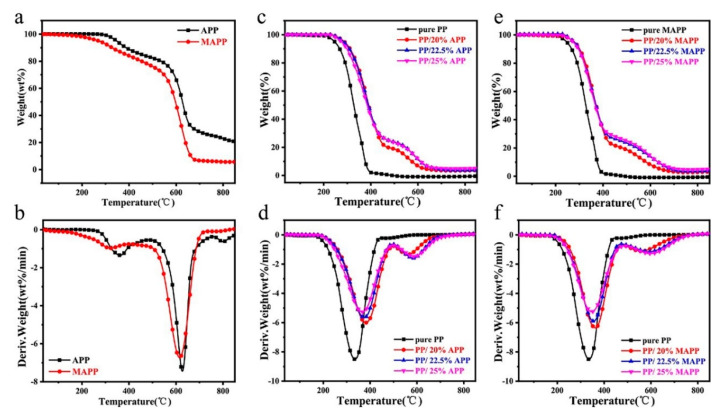
(**a**) TG and (**b**) DTG curves of APP and MAPP. (**c**,**e**) TG and (**d**,**f**) DTG. curves of PP and its flame-retardant blends under the air atmosphere.

**Figure 7 polymers-12-02761-f007:**
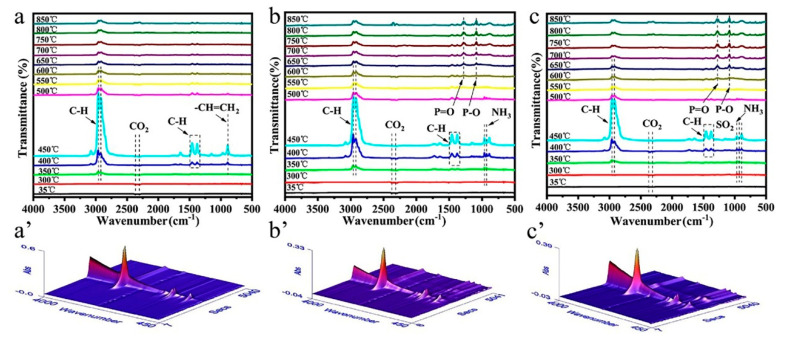
FTIR spectra and 3D images of the pyrolysis products of pure PP (**a**,**a’**), PP/25%APP (**b**,**b’**), and PP/25%MAPP (**c**,**c’**) at different temperatures.

**Figure 8 polymers-12-02761-f008:**
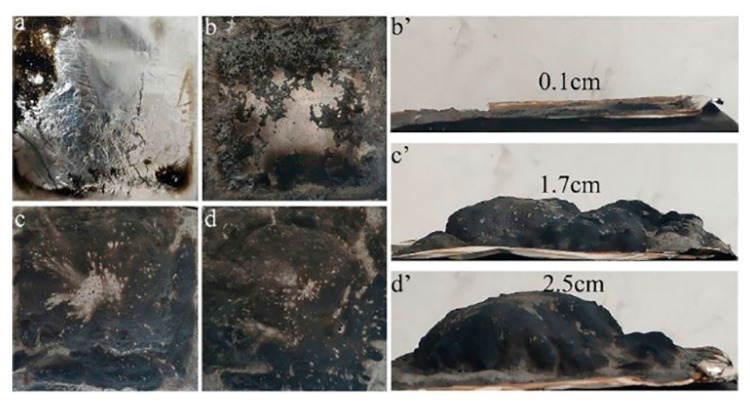
Digital photographs of residues of pure PP (**a**), PP/25%APP (**b**,**b’**), PP/22.5%MAPP (**c**,**c’**), and PP/25%MAPP (**d**,**d’**) samples after CCT.

**Figure 9 polymers-12-02761-f009:**
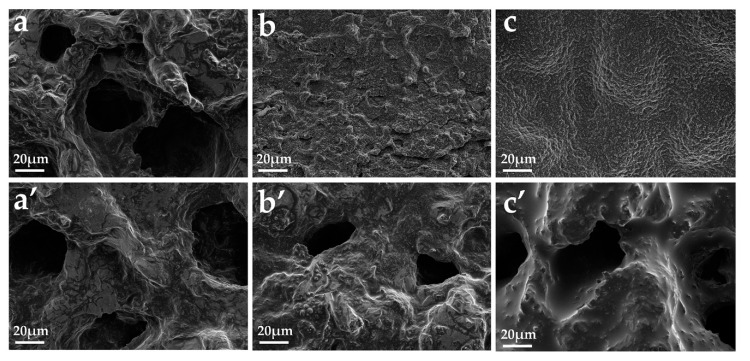
SEM images of char residues of PP/25%APP ((**a**) outer, (**a’**) inner), PP/22.5%MAPP ((**b**) outer, (**b’**) inner), and PP/25%MAPP ((**c**) outer, (**c’**) inner) after CCT.

**Figure 10 polymers-12-02761-f010:**
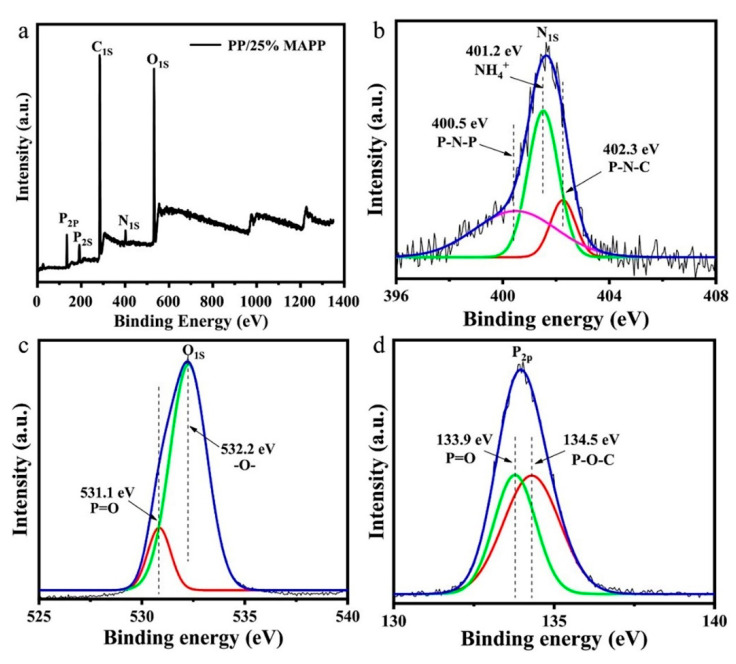
The wide-scan (**a**), N_1S_ (**b**), O_1S_ (**c**), and P_2P_ (**d**) XPS spectra of char residue of PP/25%MAPP after CCT.

**Figure 11 polymers-12-02761-f011:**
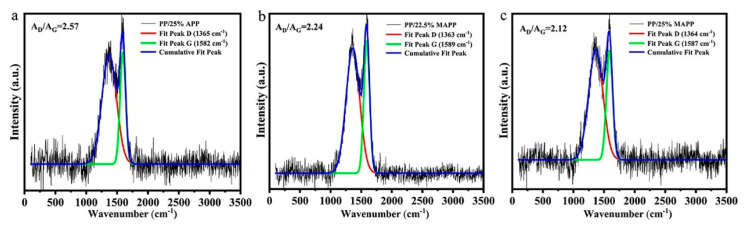
Raman spectra of char residues of (**a**) PP/25%APP, (**b**) PP/22.5%MAPP, (**c**) PP/25%MAPP after CCT.

**Figure 12 polymers-12-02761-f012:**
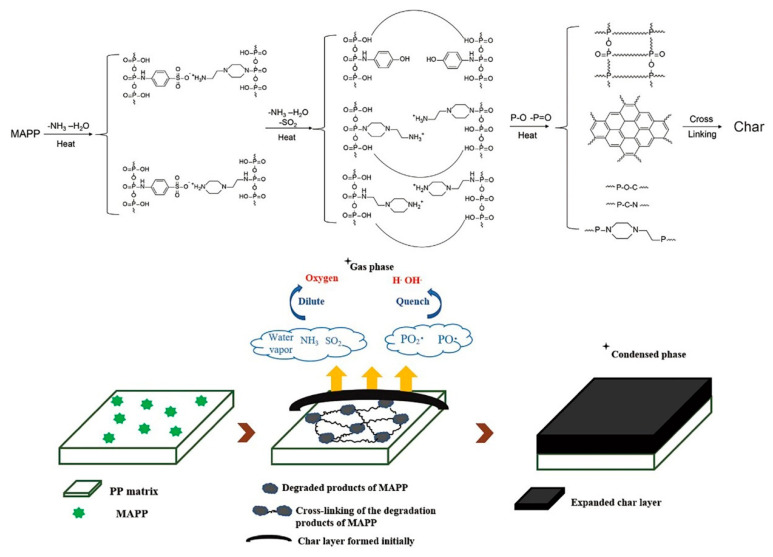
Schematic diagram of the flame-retardant mechanism.

**Figure 13 polymers-12-02761-f013:**
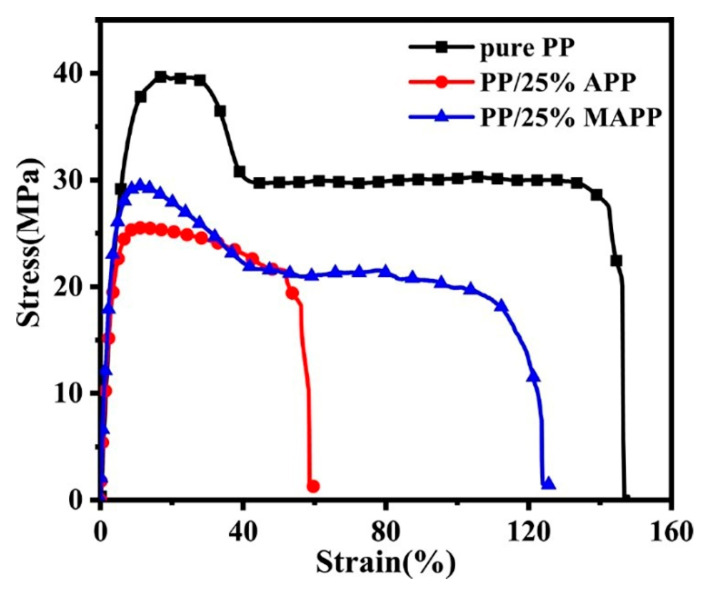
Typical stress-strain curves of pure PP and its flame-retardant blends.

**Figure 14 polymers-12-02761-f014:**
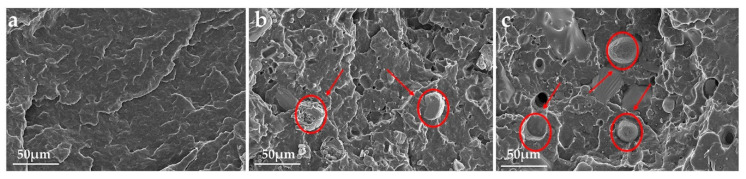
SEM images of the impact fractured surface of pure PP (**a**), PP/25%APP (**b**), and PP/25%MAPP (**c**).

**Table 1 polymers-12-02761-t001:** Detailed results for PP and its flame-retardant blends from UL-94 and LOI tests.

Sample ID	PP wt.%	APP wt.%	MAPP wt.%	Dripping or Not	UL-94 Rating 3.2 mm	LOI %
Pure PP	100	0	0	Yes	No rate	17
PP/20%APP	80	20	0	Yes	No rate	17
PP/22.5%APP	77.5	22.5	0	Yes	No rate	18
PP/25%APP	75	25	0	Yes	V-2	19
PP/20%MAPP	80	0	20	Yes	V-2	22
PP/22.5%MAPP	77.5	0	22.5	No	V-0	30
PP/25%MAPP	75	0	25	No	V-0	32

**Table 2 polymers-12-02761-t002:** Data of the pure PP and its flame-retardant blends during Cone calorimeter combustion.

Sample (Units)	Pure PP	PP/25%APP	PP/20%MAPP	PP/22.5%MAPP	PP/25%MAPP
TTI (s)	47	49	40	38	35
PHRR (kW/m^2^)	718.3	354.7	201.0	192.4	155.9
T_PHRR_ (s)	105	190	255	235	215
THR (MJ/m^2^)	57.3	47.9	50.7	47.2	44.9
TSP (m^2^)	48.8	6.4	6.5	5.2	4.2
Mean COY (kg/kg)	0.32	0.11	0.17	0.17	0.15
PSPR (m^2^/s)	0.5	0.045	0.026	0.024	0.016
AMLR (g/s)	0.059	0.043	0.033	0.032	0.027
FPI (m^2^s/kW)	0.07	0.14	0.20	0.20	0.22
FGI (kW/m^2^s)	6.84	1.87	0.79	0.82	0.72

**Table 3 polymers-12-02761-t003:** TGA data of the IFR-PP samples under nitrogen atmosphere.

Sample	*T*_5*wt*%_ (°C)	*T*_*max*1_ (°C)	*T*_*max*2_ (°C)	Rate of *T*_*max*1_ (wt.%/min)	800 °C Char Residues (%)
APP	345.1	364.4	623.9	7.3	20.5
MAPP	303.1	351.7	588.4	4.2	20.1
pure PP	373.5	455.4	/	17.7	0
PP/20%APP	387.4	448.8	687.9	8.4	8.1
PP/22.5%APP	395.9	456.6	719.1	8.2	10.9
PP/25%APP	407.9	463.7	747.4	8.1	12.5
PP/20%MAPP	383.0	454.6	704.6	8.8	8.3
PP/22.5%MAPP	368.0	446.1	694.8	8.0	8.5
PP/25%MAPP	361.2	437.6	670.9	7.6	8.6

**Table 4 polymers-12-02761-t004:** Data of TGA for PP and its flame-retardant blends under air atmosphere.

Sample	*T*_5*wt*%_ (°C)	*T*_*max*1_ (°C)	*T*_*max*2_ (°C)	Rate of *T*_*max*1_ (wt.%/min)	800 °C Char Residues (%)
APP	346.6	359.5	626.8	7.4	22.9
MAPP	270.9	320.7	615.0	6.7	5.7
pure PP	246.4	334.8	/	8.5	0
PP/20%APP	287.9	381.1	561.3	6.0	3.7
PP/22.5%APP	284.2	374.1	583.7	5.7	3.9
PP/25%APP	275.0	366.0	589.9	5.3	5.3
PP/20%MAPP	279.3	362.1	557.1	6.4	3.3
PP/22.5%MAPP	277.3	354.1	591.0	5.9	4.1
PP/25%MAPP	273.8	346.0	596.6	5.6	5.4

**Table 5 polymers-12-02761-t005:** Mechanical properties data of pure PP and its flame-retardant blends.

Sample	Tensile Strength (MPa)	Elongation at Break (%)	Impact Strength (kJ m^−2^)
Pure PP	39.1 ± 0.4	147.1 ± 3.0	4.6 ± 0.5
PP/25%APP	25.3 ± 2.5	59.8 ± 6.8	2.9 ± 0.2
PP/25%MAPP	28.4 ± 1.1	125.9 ± 4.8	4.0 ± 0.2

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
