# Peer review of "Cross-Linking Modification of Ammonium Polyphosphate via Ionic Exchange and Self-Assembly for Enhancing the Fire Safety Properties of Polypropylene"

_polymers, 2020, doi:10.3390/polym12112761_

Round 1
Reviewer 1 Report
A nice paper, scientifically sounded and describing an innovative fire retardancy strategy.
only minor changes are required prior publication:
Please improve literature citation. Plenty of paper described IFR system as made of the acid source, blowing agent, and carbonizing agent before ref. 8. please cite the basic work.
Please add charge + in Fig 1 to the NH4+ in APP (second row)
about MAPP synthesis and characterization: do N-H and -NH2 groups in the EAP react with ASC to the same extent? do the three amino groups in the precursor ehibit the same ability to exchange ion with APP? Is it possible to see a possible different rectivity of these groups by IR (primary and secondar amine salts should absorb at somewhat different vawenumbes?)
There is any explication why the crystalline structure of APP is maintained MAPP
Please explain COY (table 3) and comment on CO2P and COP in figs 4e and 4f
Reviewer 2 Report
This manuscript deals with the use of modified ammonium polyphosphate as intumescent flame retardant formulation for PP. while I found this paper interesting I think it should be revised before being accepted. Here my main comments.
- The formation of MAPP as described by the authors in figure 1 is not proven. The achieved results for XRD and IR would suggest that a simple mixture of the two components is achieved
- The authors should prepare a reference material where a simple mixture of the two components is prepared and tested. This is mandatory in order to prove the efficiency of the proposed ion exchange approach
- Tables need the error added.
- The thermal stability section should be better described, the PP degradation should also be discussed. The authors should also comment on the fact that APP containing samples and MAPP are quite similar.
